The superior response speed of table tennis players is associated with proactive inhibitory control

Zhu Mengyan 1
Pi Yanling 2
Zhang Jian 1
Gu Nan 3 gunan1@sus.edu.cn
1 School of Psychology, Shanghai University of Sport , Shanghai , China
2 Shanghai Punan Hospital of Pudong New District , Shanghai , China
3 School of Physical Education and Coaching, Shanghai University of Sport , Shanghai , China
Pazzaglia Mariella
Electronic publication date: 2022 May 20
Publication date: 2022
Volume: 10
Electronic Location ID: e13493
Received 2022 Jan 25; Accepted 2022 May 4
Copyright: © 2022 Zhu et al.
Copyright year: 2022
Copyright holder: Zhu et al.
License: This is an open access article distributed under the terms of the Creative Commons Attribution License, which permits unrestricted use, distribution, reproduction and adaptation in any medium and for any purpose provided that it is properly attributed. For attribution, the original author(s), title, publication source (PeerJ) and either DOI or URL of the article must be cited.
License URL: https://creativecommons.org/licenses/by/4.0/

Keywords: Proactive inhibitory control, Reaction time, Table tennis player, Open skill sports

Funding: National Natural Science Foundation of China 11932013 and 31971024 Natural Science Foundation of Shanghai 19ZR1453000 Outstanding Clinical Discipline Project of Shanghai Pudong PWYgy2018-04 This work was supported by the National Natural Science Foundation of China (Grant Nos. 11932013 and 31971024); the Natural Science Foundation of Shanghai (Grant No. 19ZR1453000); and the Outstanding Clinical Discipline Project of Shanghai Pudong (Grant No. PWYgy2018-04). The funders had no role in study design, data collection and analysis, decision to publish, or preparation of the manuscript.

==============================
Objective

To explore the mechanism behind the faster volitional reaction time (RT) of open skill sports athletes from the perspective of proactive inhibitory control, with the hypothesis that the superior response speed of athletes from open skill sports is related to their enhanced capacity for releasing inhibition.

Methods

Participants were divided into two groups, an experimental group of 27 table tennis players and a control group of 27 non-athletes. By manipulating cue–target onset asynchrony (CTOA) in a simple cue-target detection task, the timing of target presentation occurred in different phases of the disinhibition process. The time needed for disinhibition were compared between groups.

Results

For the experimental group, RT varied with CTOA at delays less than 200 ms; for CTOAs greater than 200 ms, RTs were not significantly different. For the control group, RT varied with CTOA for delays as long as 300 ms.

Conclusions

Table tennis players took less time (200 ms) than non-athletes (300 ms) to complete the disinhibition process, which might partly explain their rapid response speed measured in unpredictable contexts.

Significance

The study provided evidence for disinhibition speed as a new index to assess the capacity of proactive inhibitory control, and provided a new perspective to explore the superior RT of athletes from open skill sports. We also offered support for the fundamental cognitive benefits of table tennis training.

Introduction

The capacity to respond swiftly to environmental events is a defining feature of athletes from open skill sports such as table tennis, football, and boxing (Castellar et al., 2019). Previous studies in simulated sport environments showed that athletes from open skill sports had faster reaction times (RTs) than athletes from closed skill sports and non-athletes. However, it is not known whether the same advantage would be observed in non-sport RT tasks (Voss et al., 2010).

To prevent inappropriate responses to unexpected events, the human brain inhibits responses when faced with unpredictable environments. Thus, the initiation of a response requires the release of proactive inhibitory control (Criaud et al., 2016; Criaud et al., 2012; Perri, 2020). Disinhibition pathways can be either endogenous or exogenous. The endogenous pathway is used when the environment is likely to require a predetermined reaction. In this case, the proactive inhibitory control is released in advance, creating a preparatory state which can quickly process sensorimotor information to trigger automatic responses when events occur. The exogenous pathway is used when there is uncertainty about the upcoming event. In this case, proactive inhibitory control is necessary to prevent inappropriate responses, and can only be lifted after confirming that the event is indeed a target event. Reactions of the exogenous pathway are generally volitional, and slower than the automatic responses of the endogenous pathway (Favre et al., 2013) (Fig. 1).

Figure 1 Theoretical illustration of response locked and unlocked by proactive inhibitory control (Favre et al., 2013).

To prevent inappropriate automatic response, the response trigger was locked by proactive inhibition. Proactive inhibitory control can be unlocked by either endogenous or exogenous pathways. In the endogenous pathway, removal of proactive inhibition occurs under internal control, before stimulus presents, allowing a rapid automatic response to stimulus. In the exogenous pathway, disinhibition occurs only after the stimulus has been identified, triggering a relatively slower, volitional response.

The environment of open skill sports is unpredictable (Pi et al., 2019; You et al., 2018), which means the athletes have to observe their surroundings and wait for the precise timing before action (Lucia et al., 2021; Nakamoto & Mori, 2008). Hence, athletes maintain a state of locked proactive inhibitory control until it is appropriate to remove inhibition (i.e., the exogenous pathway) and their reactions to external events are mostly volitional rather than automatic. Thus, we believe that the superiority of response speed in athletes is mainly lied in volitional response. This may partly explain the athletes’ inconsistent results in tests of general RT, which may be that some studies tested volitional RTs while others tested automatic RTs. For example, Kida, Oda & Matsumura (2005) measured the RTs of baseball players by a general Go/No-go task and a simple RT task. The results showed that baseball players outperformed controls only in the Go/No-go task. Bianco et al. (2017b) found that fencers and boxers reacted faster than non-athletes in a general Go/No-go task. Abernethy (1990) and Nielsen & McGown (1985) reported that athletes from open skill sports had no obvious RT advantage in general simple RT tasks. The Go/No-go task, a classical task to study proactive inhibitory control, tested volitional RTs due to its unpredictability (Aron, 2011), while the simple RT task tested automatic RTs. Therefore, the key point to the obvious RT superiority of athletes is probably not whether they are in a specific-sport situation, but whether the response is volitional.

This study aimed to explore the mechanism behind the faster volitional RTs of athletes from open skill sports. Considering that volitional RT is affected by disinhibition process, we hypothesized that the rapid speed of disinhibition may partly explain. Although other factors could be responsible, such as motor readiness (Bianco et al., 2017a, 2017b), but few reported the disinhibition speed.

To verify the aforementioned hypothesis, we used a simple cue-target detection task to compare the disinhibition time between table tennis players and non-athletes (Boulinguez et al., 2008; Favre et al., 2013). The stimuli of the task include cue and target. The cue serves as a prompt, appearing at different time points before the target (or not at all) and requiring no response. The potential cue is both an unpredictive stimulus that impedes rapid automatic response and a predictive prompt to enable disinhibition, thus activating the exogenous pathway. The exogenous disinhibition pathway in this task begins either when the cue appears or when the target appears (in conditions when no cue is present). RTs vary with the cue-target onset asynchrony (CTOA): if CTOA is shorter than 300 ms, the RTs are relatively long; if CTOA is longer than 300 ms, the RTs are almost as short as those measured in automatic responses (Boulinguez et al., 2009). In other words, when CTOA is longer than the time needed to release proactive inhibitory control, the disinhibition process is complete before the target is presented, and RTs are not affected. When CTOA is shorter than the time needed to release proactive inhibitory control, RTs are slowed down by the inhibition. In general, The CTOA where the RT begins to approach the automatic response RT could represent the approximate time for disinhibition (Fig. 2).

Figure 2 Timeline of exogenous and endogenous disinhibition in simple cue-target detection tasks (Favre et al., 2013).

Red lines indicate proactive inhibitory control. In the simple cue-target detection task with 0% possibility of cue occurrence, the remove of proactive inhibitory control happens endogenously, the disinhibition process ends before presentation of the target, and RT is not affected (i.e., automatic RT) (G). In the simple cue-target detection task with 50% possibility of cue occurrence, the exogenous pathway could occur two different ways: if disinhibition begins at the same time as target presentation (i.e., no-cue trials), RT will be affected by the entire disinhibition process (A); if disinhibition beings at the same time as cue presentation (i.e., cue trials), effects on RT will depend on the magnitude of cue-target onset asynchrony (CTOA) (B–F). When the CTOA begins to approach the time needed for disinhibition (D), the RT of this condition is close to the automatic RT, and the CTOA represents the approximate time for disinhibition.

Table tennis, along with badminton and squash are reactive sports (Kalpesh Vidja, 2012), which means table tennis requires athletes react fast due to its high speed and constantly changing direction (Castellar et al., 2019). Previous studies have reported the RT superiority of table tennis players (Deepa & Sirdesai, 2016). You et al. (2018) found that table tennis players showed faster RTs than non-athletes in a general Go/No-go task. Therefore, we recruited table tennis players as the experimental group. We hypothesized that table tennis players outperformed the controls only in volitional RTs, and took less time when they release proactive inhibitory control.

Materials and Methods

Participants

The sample size was calculated by G*power 3.1. The effect size was 0.25 and the α was 0.05. To make the power of a statistical test to 0.95 (He et al., 2021), the study needed 26 participants totally. Actually, 29 table tennis players and 31 non-athletes were recruited. All participants were recruited from Shanghai University of Sport. All were right-handed and had normal or corrected-to-normal vision. All table tennis players were at national level 2 or above, with 10 to 15 years of training experience (average 12 years). The control group participated in sports occasionally but had no long-term experience. The study was approved by the Ethics Committee of Shanghai University of Sport (Ethics No. 102772019RT012). All participants signed the written informed consent. The data of six participants were excluded on the basis of insufficient task performance (<95% accuracy). The final groups comprised 27 table tennis players (aged 18–21 years, average 19.93; 18 female) and 27 controls (aged 18–25, average 22.41; 15 female).

Experimental procedure

The experimental task was a simple cue-target detection task. All stimuli were displayed on a 14-inch computer screen (resolution: 1366 × 768) positioned 50 cm from the subject’s eyes. E-Prime 3.0 software was used to run the task program and complete data collection.

The experimental session included two blocks of tasks: a mixed block containing a random mixture of no-cue and cue-target trials, and a baseline block containing only no-cue trials, which was used to measure the automatic response speed. In no-cue trials, a fixation point consisting of a white cross (1.03° × 1.03°) remained in the center of the screen. Participants were instructed to focus on the fixation point. After a delay of 1,100–1,600 ms the target, a white “X” (2.46° × 3.55°), presented at 3.89° randomly to the left or right of the fixation point for a duration of 50 ms. There was a 2,000-ms delay following the target presentation. In cue-target trials the cue, consisting of two white squares (4.18° × 5.14°), appeared at 3.89° to both sides of the fixation point for a duration of 50 ms. The cue would appear randomly 100, 200, 300, 400, or 500 ms earlier than the target, denoted as CTOA-100, CTOA-200, CTOA-300, CTOA-400, and CTOA-500. In both cue and no-cue trials, participants were instructed to press “↓” key on a keyboard with their right index finger as soon as they saw the target (Fig. 3).

Figure 3 Sample images from the cue-target detection task.

No-cue and cue-target trials were randomly mixed in the task. No-cue trials: 1,100–1,600 ms after the fixation point (white cross) appeared, the target (“X”) was presented randomly on the left or right side for 50 ms, then followed by a delay of 2,000 ms. Cue-target trials: the cue (two white squares) appeared on both sides of the fixation point for 50 ms, either 100, 200, 300, 400, or 500 ms before the target appeared. In both types of trials, participants were asked to press “↓” key on the keyboard with right index finger as soon as they saw the target.

The experimental procedure began with a training baseline block consisting of 40 no-cue trials followed by a training mixed block of 40 trials. Participants were instructed to comply with a maximum error rate of 5% in each block. Two types of errors were possible: (1) anticipation, defined as responding before the target appeared or having an RT shorter than 150 ms; (2) omission, defined as no response or having an RT longer than 1,000 ms. Whenever an error occurred, an error signal and the current error rate were displayed on the screen. The formal experimental task consisted of a baseline block with 80 trials and a mixed block with 360 trials. Participants were encouraged again to comply with a maximum error rate of 5% in each block, and to respond as quickly as possible. The order of the two blocks was counterbalanced across participants.

Statistical analyses

A 2 (group: table tennis players, non-athletes) × 3 (condition: no-cue/mixed, no-cue/baseline, cue-target) repeated ANOVA was performed for each type of error (anticipation and omission). Similarly, a 2 (group: table tennis players, non-athletes) × 7 (condition: no-cue/mixed, CTOA-100, CTOA-200, CTOA-300, CTOA-400, CTOA-500, no-cue/baseline) repeated ANOVA was performed for mean RTs. RTs that were more than three standard deviations from the mean were excluded from analysis. All tests were two-sided and considered significant when P < 0.05. Post-hoc comparisons were performed by Bonferroni tests.

Results

Errors

Anticipation

A significant effect of condition was observed, F(2, 104) = 173.679, P < 0.001, ŋp2 = 0.770. Post-hoc comparisons reported that the anticipation rate was smaller in no-cue/mixed trials (M = 0.000%, SD = 0.000%) and no-cue/baseline trials (M = 0.069%, SD = 0.289%) than in cue-target trials (M = 3.909%, SD = 2.150%). No significant effect was observed for group (F(1, 52) = 0.038, P = 0.847), or for the group × condition interaction (F(1, 52) = 0.107, P = 0.899).

Omission

No significant effect was observed for group (F(1, 52) = 0.099, P = 0.754), condition (F(2, 104) = 1.484, P = 0.231), or group × condition interaction (F(1, 52) = 0.159, P = 0.853).

Reaction times

No RT difference between response to the left and right stimulus was observed. There was a significant effect of group on reaction time (F(1, 52) = 4.475, P = 0.039, ŋp2 = 0.079). Post-hoc tests showed that RTs of the experimental group (M = 307.169 ms, SD = 50.124) were significantly shorter than those of the control group (M = 324.889 ms, SD = 53.371). There was also a significant effect of condition (F(6, 312) = 346.923, P < 0.001, ŋp2 = 0.870). RTs for CTOA-200 (M = 321.496 ms, SD = 38.261) were significantly shorter than those for CTOA-100 (M = 367.334 ms, SD = 38.627), and RTs for CTOA-300 (M = 291.544 ms, SD = 36.045) were significantly shorter than those for CTOA-200. A significant group × condition interaction was observed, F(6, 312) = 2.426, P = 0.026,ŋp2 = 0.045. Post-hoc comparisons showed that the RTs of the experimental group were significantly shorter than those of the control group in no-cue/mixed, CTOA-100, and CTOA-200; in the other four conditions, no significant RT differences were observed between the two groups. For the experimental group, RTs for CTOA-200 were significantly shorter than those for CTOA-100, but there was no significant difference in RTs between CTOA-300 and CTOA-200. For the control group, RTs for CTOA-300 were significantly shorter than those for CTOA-200 (Fig. 4).

Figure 4 RTs in seven conditions.

Compared with the control group, the RTs of the experimental group in no-cue/mixed, CTOA-100, and CTOA-200 trials were significantly shorter (P values = 0.003, 0.023, and 0.033, respectively). The RTs of both groups in CTOA-200 was significantly shorter than that in CTOA-100 (P < 0.001). No significant difference was observed between CTOA-300 and CTOA-200 for the experimental group (P = 0.229), but in the control group the RTs for CTOA-300 were significantly shorter than those for CTOA-200 (P = 0.003). Note: #: between-group variance, *: between-condition variance; ***: P < 0.001, ** & ##: P < 0.01, #: P < 0.05.

Discussion

In the present study, the dynamics of RTs in the control group were consistent with those reported previously. That is, the longer the delay between disinhibition and target presentation, the shorter the response time, approaching the RTs measured in automatic responses (Jaffard et al., 2007; Jaffard et al., 2008). Boulinguez et al. (2009) found that RT was no longer affected by proactive inhibitory control when CTOA reached 300 ms, thus it was speculated that the time required for untrained people to complete the disinhibition process was about 300 ms. Here we found that the RTs of the experimental group were already approaching those measured in automatic responses when CTOA was as low as 200 ms, while the RTs of the control group did not approach those of automatic responses until the CTOA reached 300 ms. Thus, we conclude that it required about 200 ms for the experimental group, but 300 ms for the control group, to completely release proactive inhibitory control. In addition, when the time interval between disinhibition and target presentation was shorter than 300 ms (no-cue/mixed, CTOA-100, and CTOA-200), a significant difference in RT was observed between groups. Conversely, when the time interval exceeded 300 ms (CTOA-300, CTOA-400, CTOA-500, and no-cue/baseline), there was no significant difference in RT between groups. Perhaps because only in these conditions (no-cue/mixed, CTOA-100, and CTOA-200) were RTs affected by proactive inhibitory control. If the experimental group required less time to lift proactive inhibitory control, their RTs would be less affected by the disinhibition process. Therefore, in conditions under the influence of proactive inhibitory control (i.e., volitional response), the experimental group showed an obvious advantage in response speed, whereas no significant advantage was observed in conditions divorced from the effect of proactive inhibitory control (i.e., automatic response).

All in all, we proved that the RT advantage of table tennis players mainly lied in the volitional response; we found that the disinhibition time of table tennis players was shorter than that of the controls, supporting that table tennis players disinhibit faster than non-athletes (Fig. 5B). It is worth noting that researchers reported athletes’ stronger proactive inhibitory control (i.e., the ERP component of prefrontal negativity, pN) (Bianco et al., 2017a, 2017b), which helped us rule out the possibility that the lower degree of inhibition leaded to the shorter time of disinhibition (Fig. 5A). The stronger proactive inhibitory control guarantees athletes’ accuracy (Bianco et al., 2017a, 2017b; Messel et al., 2019; Xu et al., 2015), which is as important as response speed for athletes from open skill sports.

Figure 5 Schematic illustration of the hypothetical model of disinhibition curves.

(A) There is no significant difference between groups in disinhibition speed, but the non-athletes have stronger proactive inhibitory control, so they need more time for disinhibition. (B) Table tennis players have stronger proactive inhibitory control, but they disinhibit faster than non-athletes, so they spend less time for disinhibition.

Inhibitory control is one important aspect of cognitive control, and can be either proactive or reactive (Braver, 2012). Proactive inhibitory control refers to the ability to control one’s attention, behavior, thought, and/or emotion; to overcome strong internal predisposition or external temptation; to refrain from reacting automatically to unpredictable events; and to instead do what’s more appropriate or needed (Braver, 2012; Diamond, 2013). It is thought to be a top-down process defined as the ability to formulate a response strategy based on the environment (e.g., slowing down while driving in a school zone) (Ballanger, 2009; Di Caprio et al., 2020; Duque et al., 2017). Reactive inhibitory control refers to the ability to withdraw from already-planned actions when necessary, and is thought to be a bottom-up, late correction process triggered by external signals (e.g., braking when a pedestrian suddenly crosses the street) (Aron, 2011; Aron, Robbins & Poldrack, 2004, 2014; Braver, 2012; Hampshire & Sharp, 2015). Previous studies found that athletes, especially those in open skill sports, exhibited a heightened capacity for inhibitory control (Huijgen et al., 2015; Jacobson & Matthaeus, 2014; Meng et al., 2019; Yamashiro et al., 2015); however, most of the studies looked at reactive rather than proactive inhibitory control (Heppe & Zentgraf, 2019; Liao, Meng & Chen, 2017; Wang et al., 2013). Using a modified version of the stop-signal task, Brevers et al. (2018) estimated proactive inhibitory control by increased go-signal RT as a function of increased stop-signal probability (i.e., a higher change in go-signal RTs per stop-signal probability unit indicated a better capacity for proactive inhibitory control) and reported that elite athletes exhibited superior proactive inhibitory control. In light of these results, we posit that proactive inhibitory control facilitates inhibition efficiency (i.e., accuracy), and measures of inhibition efficiency can provide an important index to a person’s ability for proactive inhibitory control. However, according to our results, we also believe disinhibition speed should be another index, which determines whether a person can react rapidly even in unpredictable circumstances.

The brain network underlying proactive inhibitory control includes right inferior prefrontal gyrus, pre-supplementary motor area, subthalamic nucleus, and striatum (Coxon, Stinear & Byblow, 2006; Meyer & Bucci, 2016). We speculated that table tennis players could release proactive inhibitory control rapidly, perhaps because they could switch swiftly between the inhibitory state and the automatic response state (Favre et al., 2013; Messel et al., 2019), a process which may be controlled by connections between this inhibition network and the primary motor cortex (Duque et al., 2017). Experience-based neuroplasticity studies have shown that repeated and frequent participation in a particular sport alters the relevant neural representations (Calvo-Merino et al., 2006; Koeneke et al., 2004; Naito & Hirose, 2014; Weisberg, van Turennout & Martin, 2007). Changes in neuroplasticity caused by sports training could explain why elite athletes outperform novices and non-athletes (Jellinger, 2007; Kelly & Garavan, 2005; Pascual-Leone et al., 2005; Raz & Lindenberger, 2013). The rapid disinhibition speed of table tennis players may be one of the manifestations of the changes in neuroplasticity caused by long-term sports training.

We must admit the limitations of this study. First, the effect of different levels of physical activity and aerobic fitness on brain and cognition could not be excluded (Erickson, Hillman & Kramer, 2015) for we could not control the sports training time of table tennis players. Second, only table tennis players were recruited. We are cautious about whether these results could be applied to athletes from other open skill sports due to different technical, tactical and strategic demands of different open skill sports, which might influence the motor-skill experience in different ways as an effect of neuroplasticity (Voss et al., 2010). Thus, athletes from other open skill sports and closed skill sports are needed to verify our results. Third, further replications of our tasks coupled with neurophysiological measurements are needed for the exploration of the disinhibition process, which is what we are going to do in future studies.

Conclusions

In conclusion, our study took table tennis players as the experimental group and, through simple cue-target detection tasks, showed that the rapid speed of releasing proactive inhibitory control might partly explain their superior performance in volitional RTs.

Supplemental Information

Supplemental Information 1 RTs of each participant in different conditions.

Click here for additional data file.

Additional Information and Declarations

Competing Interests

Author Contributions

Human Ethics

Data Availability

The authors declare that they have no competing interests.

Mengyan Zhu conceived and designed the experiments, performed the experiments, analyzed the data, prepared figures and/or tables, authored or reviewed drafts of the paper, and approved the final draft.

Yanling Pi conceived and designed the experiments, authored or reviewed drafts of the paper, and approved the final draft.

Jian Zhang conceived and designed the experiments, analyzed the data, authored or reviewed drafts of the paper, and approved the final draft.

Nan Gu conceived and designed the experiments, authored or reviewed drafts of the paper, and approved the final draft.

The following information was supplied relating to ethical approvals (i.e., approving body and any reference numbers):

The study was approved by the Ethics Committee of Shanghai University of Sport (Ethics No. 102772019RT012).

The following information was supplied regarding data availability:

The raw data are available in the Supplemental File.

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
