# Peer review of "The superior response speed of table tennis players is associated with proactive inhibitory control"

_PeerJ, doi:10.7717/peerj.13493_

## Round 0.1 · original submission · Major Revisions

Dear authors,

As usual, I tried to get two experts from your research area on board.

On the one hand, this is a very interesting approach and study. On the other hand, however, too many questions remain. As you will read, the reviewers note the potential interest of your paper, but the reviewers also raise a number of concerns about the method and the interpretation of your results that need to be addressed.

Reviewer 1 ·

Basic reporting

The captions on figure 1 and 2 may be too small for the final print.

Experimental design

no comment

Validity of the findings

no comment

Additional comments

Authors start with the observation that open skill athletes have faster RT in the sport specific
manner. Previous studies fail to show whether this superior performance can be generalized
to other non-sport contexts. Here, authors came up with the hypothesis that the simple RT
task cannot account for the faster response of athletes because of their supremacy can shine
in unpredictable environments. Then, what’s so special about this unpredictable sport
situations? Authors suggests, being able to block un-desirable responses when many options
are on the table may be a contributing factor for superior response speed of athletes. Thus,
as a way investigate this proactive inhibitory control ability, the CTOA paradigm (a warning
signal with variable SOA) during the RT test was adopted.

The main finding of this study showed that the RTs of the experimental group were
significantly shorter than those of the control group in no-cue/mixed, CTOA-100, and CTOA-
200 conditions. This selective difference in proactive inhibitory control seems to explain the
characteristics of athletes described in the introduction section, and provide an evidence that
this characteristic is observable in non-sports specific context supporting the hypothesis. The
fact that performances requiring reactive inhibitory control didn’t show group differences re-
confirms the results are proactive inhibitory control specific. Finally, possible cognitive
mechanisms that may contribute such group difference was suggested.

In short, this manuscript is well prepared, and the experiments are well prepared/executed.
The results are almost exactly what the hypothesis expected – therefore there is not much
room for discussion. I believe the manuscript is ready to publish without much revision.

Here are some very minor concerns that authors may consider reflecting.

- The overall hypothesis & discussion are heavily rooted on the assumption that the
CTOA experimental paradigm is the index of proactive disinhibition control. Thus, it
may be useful to provide some more details regarding the nature of disinhibition in this
task. In this sense, Fig 1 and 2 is useful but some more written descriptions would be
informative, too.

- At times, the expressions are not quite natural, or have vague expressions.

- Did the subjects respond to either Left or Right target with one finger press movement
option? If so, was there any RT difference between those stimuli?

- Since this study was about the superior ability of athletes, a little more discussion
regarding the reason for such group difference would be beneficial. In the abstract,
authors mentioned this may due to the training or experience. You may elaborate this
point.

Annotated reviews are not available for download in order to protect the identity of reviewers who chose to remain anonymous.

Reviewer 2 ·

Basic reporting

-Authors are encouraged to cite and discuss more recent studies which have already explored the relation between RT and proactive inhibitory control. (e.g. Bianco, V., Di Russo, F., Perri, R. L., & Berchicci, M. (2017). Different proactive and reactive action control in fencers’ and boxers’ brain. Neuroscience, 343, 260-268.; Bianco, V., Berchicci, M., Perri, R. L., Quinzi, F., & Di Russo, F. (2017). Exercise-related cognitive effects on sensory-motor control in athletes and drummers compared to non-athletes and other musicians. Neuroscience, 360, 39-47.)
-In the introduction section, authors claim that they chose table tennis player as the most representative open-skill sport… this is not true, since also volleyball, tennis, fencing and boxing are for instance appropriate open-skill sports… therefore, authors are encouraged to expand this literature with recent studies e.g. (Montuori, S., D'Aurizio, G., Foti, F., Liparoti, M., Lardone, A., Pesoli, M., ... & Sorrentino, P. (2019). Executive functioning profiles in elite volleyball athletes: Preliminary results by a sport-specific task switching protocol. Human movement science, 63, 73-81; Lucia, S., Bianco, V., Boccacci, L., & Di Russo, F. (2022). Effects of a Cognitive-Motor Training on Anticipatory Brain Functions and Sport Performance in Semi-Elite Basketball Players. Brain Sciences, 12(1), 68.; Koch, P., & Krenn, B. (2021). Executive functions in elite athletes–Comparing open-skill and closed-skill sports and considering the role of athletes' past involvement in both sport categories. Psychology of Sport and Exercise, 55, 101925.).
-Accordingly, authors clearly state why they the selected table tennis players instead of other open-skill sports.
-It is not clear to me the reasons behind the choice of a cue–target onset asynchrony (CTOA) to study proactive inhibitory control. Authors should acknowledge other tasks commonly used to investigate proactive inhibitory control as for instance the go/nogo or the stop signal task (see he already cited Aron, A. R. (2011). From reactive to proactive and selective control: developing a richer model for stopping inappropriate responses. Biological psychiatry, 69(12), e55-e68.)

Experimental design

-Authors should provide an accurate power analysis which justify that the sample size is appropriate for the RM-ANOVA design of the study
-Why the authors did not measure physical activity level? How can we be fully sure that the effects between athletes and non-athletes are not a consequence of different physical activity levels (Erickson, K. I., Hillman, C. H., & Kramer, A. F. (2015). Physical activity, brain, and cognition. Current opinion in behavioral sciences, 4, 27-32.)? I believe that this might represent a limitation of the study since the levels of physical activity and aerobic fitness were not even estimated.
- It is not clear how this behavioral task performed by table-tennis players filled the gap in literature. Accordingly, one other main limitation is the absence of a closed skill group or of a different open-skill sport to compare the results with.

Validity of the findings

-Authors should acknowledge the limitations of the study in a dedicated paragraph.
-Open skill sports include many sports with huge differences in technical, tactical and strategic demands. This might influence the motor-skill experience in different ways as an effect of neuroplasticity (Voss, M. W., Kramer, A. F., Basak, C., Prakash, R. S., & Roberts, B. (2010). Are expert athletes ‘expert’in the cognitive laboratory? A meta‐analytic review of cognition and sport expertise. Applied cognitive psychology, 24(6), 812-826.). Could it be more appropriate to properly refer to the participants of the present study? And then it would seem correct to justify results in terms of open skill experience. Therefore, I would suggest to mitigate in extending present finding to all open skills.
-Being a behavioral study, the authors might suggest further replications of the task coupled with neurophysiological measurements.

---

## Round 0.2 · accepted · Accept

Thank you for your interest in submitting your work to PeerJ
Congratulations